# Impact of Changes in Serum Calcium Levels on In-Hospital Mortality

**DOI:** 10.3390/medicina56030106

**Published:** 2020-03-02

**Authors:** Charat Thongprayoon, Wisit Cheungpasitporn, Panupong Hansrivijit, Juan Medaura, Api Chewcharat, Michael A Mao, Tarun Bathini, Saraschandra Vallabhajosyula, Sorkko Thirunavukkarasu, Stephen B. Erickson

**Affiliations:** 1Division of Nephrology and Hypertension, Department of Medicine, Mayo Clinic, Rochester, MN 55905, USA; api.che@hotmail.com (A.C.); Thirunavukkarasu.Sorkko@mayo.edu (S.T.); Erickson.Stephen@mayo.edu (S.B.E.); 2Division of Nephrology, Department of Internal Medicine, University of Mississippi Medical Center, Jackson, MS 39216, USA; jmedaura@umc.edu; 3Department of Internal Medicine, University of Pittsburgh Medical Center Pinnacle, Harrisburg, PA 17101, USA; p.hansrivijit@gmail.com; 4Division of Nephrology and Hypertension, Mayo Clinic, Jacksonville, FL 32224, USA; mao.michael@mayo.edu; 5Department of Internal Medicine, University of Arizona, Tucson, AZ 85721, USA; tarunjacobb@gmail.com; 6Department of Cardiovascular Medicine, Mayo Clinic, Rochester, MN 55905, USA; Vallabhajosyula.Saraschandra@mayo.edu

**Keywords:** calcium, electrolytes, hypocalcemia, hypercalcemia, mortality

## Abstract

*Background and objectives:* Calcium concentration is strictly regulated at both the cellular and systemic level, and changes in serum calcium levels can alter various physiological functions in various organs. This study aimed to assess the association between changes in calcium levels during hospitalization and mortality. *Materials and Methods:* We searched our patient database to identify all adult patients admitted to our hospital from January 1st, 2009 to December 31st, 2013. Patients with ≥2 serum calcium measurements during the hospitalization were included. The serum calcium changes during the hospitalization, defined as the absolute difference between the maximum and the minimum calcium levels, were categorized into five groups: 0–0.4, 0.5–0.9, 1.0–1.4, 1.5–1.9, and ≥2.0 mg/dL. Multivariable logistic regression was performed to assess the independent association between calcium changes and in-hospital mortality, using the change in calcium category of 0–0.4 mg/dL as the reference group. *Results:* Of 9868 patients included in analysis, 540 (5.4%) died during hospitalization. The in-hospital mortality progressively increased with higher calcium changes, from 3.4% in the group of 0–0.4 mg/dL to 14.5% in the group of ≥2.0 mg/dL (*p* < 0.001). When adjusted for age, sex, race, principal diagnosis, comorbidity, kidney function, acute kidney injury, number of measurements of serum calcium, and hospital length of stay, the serum calcium changes of 1.0–1.4, 1.5–1.9, and ≥2.0 mg/dL were significantly associated with increased in-hospital mortality with odds ratio (OR) of 1.55 (95% confidence interval (CI) 1.15–2.10), 1.90 (95% CI 1.32–2.74), and 3.23 (95% CI 2.39–4.38), respectively. The association remained statistically significant when further adjusted for either the lowest or highest serum calcium. *Conclusion:* Larger serum calcium changes in hospitalized patients were progressively associated with increased in-hospital mortality.

## 1. Introduction

Calcium is the most abundant mineral in the human body and has many essential functions including muscle function, nerve transmission, intracellular signaling, and mediating vascular contraction and vasodilatation [1,2]. Furthermore, cardiac contractility is also regulated by changes in intracellular calcium concentration [3,4]. Thus, calcium level is strictly regulated at both the cellular and systemic levels [5,6,7]. 

Serum calcium levels are affected by several important factors including vitamin D and parathyroid hormone (PTH), serum phosphate, and magnesium levels [5,8,9,10]. Abnormalities of calcium derangements and changes in serum calcium levels are common in clinical practice, especially among critically ill patients with prevalence up to 88% [5,10,11,12,13,14,15]. While a decrease in serum calcium is associated with severity of illness [5,11,12], an increase in serum calcium among hospitalized patients is commonly seen in primary hyperparathyroidism, certain medications, and diseases, such as underlying malignancy [16,17,18]. Previous studies have demonstrated the impact of serum calcium levels, including hypocalcemia and hypercalcemia, on poor clinical outcomes including mortality among hospitalization [15,19,20]. However, little is known about the effects of changes in serum calcium level on outcomes during hospitalization. 

Thus, we conducted this study to evaluate the relationship between changes in serum calcium level during hospitalization and mortality in all hospitalized patients.

## 2. Materials and Methods

### 2.1. Study Population

We searched our patient database to identify all adult patients admitted to our hospital from January 1st, 2009 to December 31st, 2013. Patients with at least 2 serum calcium measurements during hospitalization were included. Patients without research authorization were excluded. If patients had recurring admissions, only the first hospitalization during the study period was included in the analysis. This study was reviewed and approved by the Mayo Clinic institutional review board (IRB number: 15-000024; Approval Date: 2/4/2015). The informed consent was waived due to the minimal risk nature of the study.

### 2.2. Data Collection

Clinical information and laboratory data were obtained from our institutional electronic health record system using the Mayo Clinic Life Science System Database. This database contains demographic characteristics, hospital admission information, diagnosis and procedure codes, laboratory test results, and flow sheet data of inpatient admission. Serum calcium change during hospitalization, defined as the absolute difference between the maximum and minimum serum calcium values during hospitalization, was the exposure of interest. Serum calcium change was categorized into 5 groups; 0–0.4, 0.5–0.9, 1.0–1.4, 1.5–1.9, ≥2.0 mg/dL.

To further assess if the direction of serum calcium change was associated with patient outcomes, the timing of the highest serum calcium in relation to the lowest serum calcium was determined. If the highest serum calcium occurred before the lowest serum calcium, a downward trend of serum calcium change was assumed, and negative values for serum calcium change (the lowest to the highest serum calcium) was assigned. In contrast, if the highest serum calcium occurred after the lowest serum calcium, an upward trend of serum calcium change was assumed, and positive values for serum calcium change (the highest to the lowest serum calcium) were assigned. Phosphate change with direction of change was categorized into 10 groups; ≤−2.0, −1.9 to −1.5, −1.4 to −1.0, −0.9 to −0.5, −0.4 to 0.0, 0.1 to 0.4, 0.5 to 0.9, 1.0 to 1.4, 1.5 to 1.9, and ≥2.0 mg/dL.

Principal diagnoses were categorized based on International Classification of Diseases, 9th Revision, (ICD-9) codes. The Charlson Comorbidity Score [21] was calculated to assess comorbid conditions at the time of admission. The medical comorbid conditions were collected utilizing a previously validated data abstraction algorithm. Estimated glomerular filtration rate (eGFR) was calculated using the Chronic Kidney Disease Epidemiology Collaboration (CKD-EPI) equation [22]. Acute kidney injury (AKI) was defined as an increase in serum creatinine of ≥0.3 mg/dL or ≥1.5 times from the most recent outpatient serum creatinine within 1 year before admission [23].

### 2.3. Clinical Outcomes

In-hospital mortality was the outcome of interest. Death status was obtained from the institutional database. 

### 2.4. Statistical Analysis

Continuous variables were presented as mean ± standard deviation (SD) or median with interquartile range (IQR), as appropriate. The differences in continuous variables between calcium change groups were tested using analysis of variance (ANOVA). Categorical variables were presented as count with percentage. The differences in categorical variables between calcium change groups were tested using Chi-squared test. The serum calcium change group of 0–0.6 mg/dL was chosen as the reference group for outcome comparison. Logistic regression analysis was conducted to assess the association between serum calcium change during hospitalization and in-hospital mortality. A multivariable model was built to adjust for priori-defined variables. Model 1 was unadjusted; model 2 was adjusted for age, race, sex, principal diagnosis, Charlson Comorbidity Score, history of coronary artery disease, congestive heart failure, peripheral artery disease, stroke, diabetes mellitus, chronic obstructive pulmonary disease, cirrhosis, eGFR, intensive care unit (ICU) admission, AKI occurrence in hospital, the number of serum calcium measurements during hospitalization, and hospital length of stay; model 3 was adjusted for all variables in model 2, plus the lowest serum calcium during hospitalization; model 4 was adjusted for all variables in model 2, plus the highest calcium during hospitalization, and model 5 was adjusted for all variables in model 2, plus the admission calcium. Two-tailed *p* value <0.05 was considered statistically significant. All analyses were performed using JMP statistical software (Version 10; SAS Institute Inc., Cary, NC, USA).

## 3. Results

### 3.1. Clinical Characteristics

A total of 9868 patients were analyzed. Fifty-five percent were male. The mean age was 61 ± 17 years. The median number of serum calcium measurements during hospitalization was 3 (2–5), and length of hospital stay was 7 (4–14) days. Given the normal reference range of total serum calcium of 8.6–10.0 mg/dL in our hospital, 1501 (15%) had elevated serum calcium above 10.0 mg/dL, and 6481 (65%) had decreased serum calcium below 8.6 mg/dL during the hospitalization. The mean serum calcium change during hospitalization was 1.0 ± 1.2 mg/dL. The clinical characteristics of patients based on serum calcium change groups are summarized in Table 1. 

### 3.2. Serum Calcium Change and In-Hospital Mortality

Out of 9868 patients included in the study, 540 (5.5%) died in hospital. The restricted cubic spline showed the U-shaped association between the direction of serum calcium change and in-hospital mortality (Figure 1). The in-hospital mortality progressively increased with higher calcium change, from 3.4% in the group of 0–0.4 mg/dL to 14.5% in the group of ≥2.0 mg/dL (*p* < 0.001) (Table 2). When adjusting for potential confounders (model 2), serum calcium changes of 1.0–1.4, 1.5–1.9, ≥2.0 mg/dL were significantly associated with increased in-hospital mortality with adjusted odds ratio (OR) of 1.55 (95% CI 1.15–2.10), 1.90 (95% CI 1.32–2.74), and 3.23 (95% CI 2.39–4.38), respectively, when compared to the serum calcium change group of 0-0.6 mg/dL. The association remained statistically significant with a serum calcium change of ≥1.0 mg/dL when further adjusting for either the lowest (model 3), or highest serum calcium (model 4). When further adjusting for admission serum calcium (model 5), the serum calcium change of ≥1.5 mg/dL was significantly associated with increases in-hospital mortality. 

### 3.3. Direction of Serum Calcium Change and In-Hospital Mortality

The higher in-hospital mortality was observed in both upward and downward trends of serum calcium change (Table 3 and Figure 1). Regarding the downward trend of serum calcium change, serum calcium change of ≤−2.0 mg/dL was significantly associated with increased in-hospital mortality. Regarding the upward trend of serum calcium change, serum calcium changes of 1.0 to 1.4, 1.5 to 1.9, and ≥2.0 mg/dL were significantly associated with increased in-hospital mortality. The OR of in-hospital mortality associated with a markedly upward trend of serum calcium change was higher than the OR of in-hospital mortality associated with a markedly downward trend of serum calcium change.

## 4. Discussion

In this study, we demonstrated that the changes in serum calcium level during hospitalization were associated with hospital mortality. Patients with changes in serum calcium level >1.0 mg/dL were associated with increased risk of mortality, while the highest mortality was observed among patients with calcium change ≥ 2 mg/dL during hospitalization. We observed the higher in-hospital mortality in both upward (≥1.0 mg/dL) and downward (≤−2.0 mg/dL) trends of serum calcium change.

Previous studies have demonstrated the impacts of admission hypocalcemia and hypercalcemia on hospital mortality [15,19]. In this study, we demonstrated adverse effects of changes in serum calcium during hospitalization on mortality, while patients with stable serum calcium level during hospitalization had the lowest hospital mortality. Decrease in serum calcium is common among critically ill patients and correlates with severity of illness [5,11,12]. Furthermore, reduction in serum calcium can occur after blood transfusions, plasma exchanges, and parathyroidectomy [24,25,26]. In this study, we comprehensively adjusted for all identifiable factors including comorbidities, and revealed a significant association of decrease in serum calcium (≤−2.0 mg/dL) during hospitalization with higher in-hospital mortality. Although the underlying mechanisms remain unclear, it is suggested that decrease in serum calcium can significantly alter the myocardium action potential and reduce renal sodium excretion resulting in fluid overload and reduced contractility [10,27,28,29,30,31]. 

Our study demonstrated that the risk of in-hospital mortality is markedly high with an upward trend of serum calcium change than the risk of in-hospital mortality in patients with a downward trend of serum calcium change. An increase in serum calcium level is frequently seen among hospitalized patients with principal diagnoses of hematologic/oncological diseases and endocrine/metabolic disorders [15,19,32]. A rise in serum calcium may enhance atherogenesis through vascular calcification and increased coagulability [33]. An increase in serum calcium is associated with incident heart failure [33] and possible hypercalcemia-induced neuronal injury among patients with acute ischemic stroke [34]. Moreover, an increase in serum calcium may result in AKI via renal vasoconstriction and nephrogenic diabetes insipidus-induced volume depletion [35,36,37,38,39,40]. In our current study, we also adjusted AKI during hospitalization in our multivariable logistic regression analysis, and we found that the upward trend of serum calcium change of >1.0 mg/dL was significantly associated with increased in-hospital mortality, while the highest mortality (6.3-fold increased mortality) was observed in patients with an absolute increase in serum calcium level ≥2.0 mg/dL.

There are some limitations to our study. First, this is a single-center retrospective cohort study conducted in a predominantly Caucasian population. This potentially limited generalizability of the study. Due to the observational nature of the study, the causal association cannot be established. In addition, the data from this study were retrieved from institutional database. Unfortunately, some clinical information such as treatment of abnormalities of calcium derangements, oral calcium supplement, intravenous calcium infusion, complete data on other electrolytes [41,42,43,44,45,46], and medications was not available in our database and, therefore, was not reported in this study. Although we extensively adjusted for potential confounders, the association between serum calcium change and mortality might remain be confounded by unmeasured confounders. Future study with a more diversified population and more comprehensive clinical information is needed to better assess the association. Second, we did not use albumin-corrected serum calcium in the analysis because of limited availability of serum albumin at the time of serum calcium measurement. Lastly, our study only addresses the impact of magnitude and direction of serum calcium change on mortality; other aspects of serum calcium change such as acuity and variability were not assessed. These factors are also important for the understanding of the impact of serum calcium change and patient outcomes.

## 5. Conclusions

In conclusion, we demonstrated that the absolute change of serum calcium during admission is associated with increased in-hospital mortality. The association progressively increases with the degree of serum calcium change. A downward trend of serum calcium change (≤−2.0 mg/dL) and an upward trend of serum calcium change ≥1.0 mg/dL were significantly associated with increased in-hospital mortality.

## Figures and Tables

**Figure 1 medicina-56-00106-f001:**
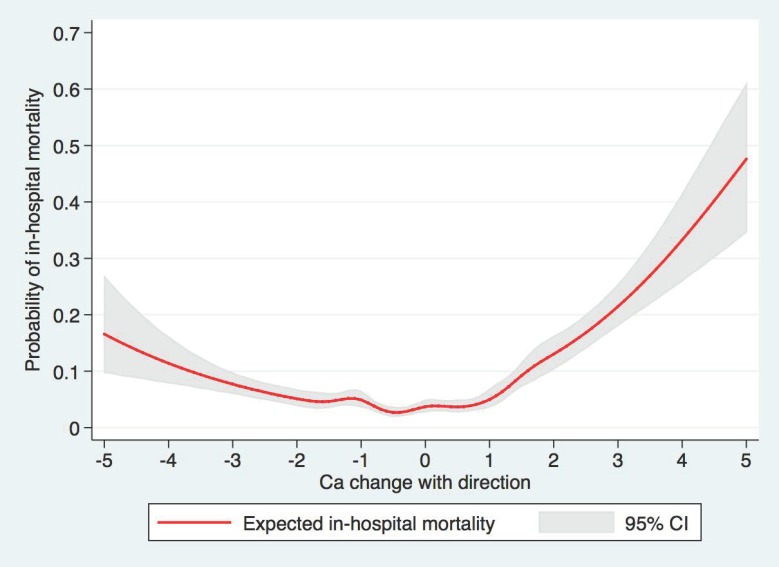
The restricted cubic spline showed the U-shaped association between the direction of serum calcium change and in-hospital mortality. CI = confidence interval.

**Table 1 medicina-56-00106-t001:** Clinical characteristics.

Variables	All	Change in Serum Calcium Level during Hospitalization (mg/dL)					
		0–0.4	0.5–0.9	1.0–1.4	1.5–1.9	≥2.0	*p* Value
**N**	9868	3200	3030	1575	817	1246	
Age (year)	61 ± 17	63 ± 17	61 ± 17	60 ± 16	58 ± 16	60 ± 16	<0.001
Male sex	5398 (55)	1808 (57)	1664 (55)	881 (56)	429 (53)	616 (49)	<0.001
Caucasian	8841 (90)	2892 (90)	2731 (90)	1405 (89)	720 (88)	1093 (88)	0.04
Principal diagnosis							<0.001
- Cardiovascular	1685 (17)	648 (20)	510 (17)	220 (14)	103 (13)	204 (16)	
- Hematology/oncology	1681 (17)	516 (16)	489 (16)	265 (17)	155 (19)	256 (21)	
- Infectious disease	675 (7)	205 (6)	183 (6)	121 (8)	43 (5)	123 (10)	
- Endocrine/metabolic	582 (6)	160 (5)	151 (5)	91 (6)	46 (6)	134 (11)	
- Respiratory	463 (5)	174 (5)	161 (5)	67 (4)	28 (3)	33 (3)	
- Gastrointestinal	1301 (13)	445 (14)	438 (14)	198 (13)	105 (13)	115 (9)	
- Genitourinary	797 (8)	206 (6)	200 (7)	131 (8)	106 (13)	154 (12)	
- Injury and poisoning	1750 (18)	496 (16)	593 (20)	341 (22)	163 (20)	157 (13)	
- Other	934 (9)	350 (11)	305 (10)	141 (9)	68 (8)	70 (6)	
Charlson comorbidity score	2.5 ± 2.7	2.6 ± 2.8	2.5 ± 2.7	2.6 ± 2.7	2.4 ± 2.6	2.4 ± 2.5	0.22
Comorbidity							
- Coronary artery disease	2147 (22)	788 (25)	680 (22)	308 (20)	142 (17)	229 (18)	<0.001
- Congestive heart failure	955 (10)	363 (11)	281 (9)	142 (9)	61 (7)	108 (9)	0.002
- Peripheral artery disease	410 (4)	165 (5)	123 (4)	52 (3)	30 (4)	40 (3)	0.007
- Stroke	796 (8)	279 (9)	277 (9)	102 (6)	51 (6)	87 (7)	0.002
- Diabetes mellitus	2542 (26)	858 (27)	793 (26)	380 (24)	183 (22)	328 (26)	0.05
- Chronic obstructive pulmonary disease	972 (10)	352 (11)	308 (10)	142 (9)	57 (7)	113 (9)	0.005
- Cirrhosis	677 (7)	200 (6)	211 (7)	122 (8)	53 (6)	91 (7)	0.35
eGFR (mL/min/1.73 m^2^)	61 ± 37	65 ± 36	65 ± 37	63 ± 38	57 ± 39	46 ± 34	<0.001
Acute kidney injury	5283 (54)	1529 (48)	1476 (49)	870 (55)	470 (58)	938 (75)	<0.001
ICU admission during hospitalization	4013 (41)	1192 (37)	1100 (36)	652 (41)	356 (44)	713 (57)	<0.001
Number of serum calcium measurements during hospitalization	3 (2–5)	2 (2–3)	3 (2–4)	4 (3–7)	5 (3–9)	8 (5–14)	<.001
Length of hospital stay (day)	7 (4–14)	6 (3–11)	7 (4–12)	8 (4–16)	9 (4–21)	12 (5–27)	<0.001
The lowest calcium	8.3 ± 0.9	8.5 ± 0.7	8.3 ± 0.7	8.1 ± 0.8	7.9 ± 1.0	7.8 ± 1.3	<0.001
The highest calcium	9.3 ± 1.3	8.8 ± 0.7	9.0 ± 0.7	9.3 ± 0.8	9.6 ± 1.1	11.2 ± 2.1	<0.001
The admission calcium	9.0 ± 1.3	8.7 ± 0.7	8.8 ± 0.8	8.9 ± 0.9	9.1 ± 1.2	9.9±2.5	<0.001

Continuous data are presented as mean ± SD; categorical data are presented as count (%); eGFR, estimated glomerular filtration rate; ICU, intensive care unit.

**Table 2 medicina-56-00106-t002:** The association between serum calcium change and in-hospital mortality.

Outcome	Change in Serum Calcium Level during Hospitalization (mg/dL)				
	0–0.4	0.5–0.9	1.0–1.4	1.5–1.9	≥2.0
N	3200	3030	1575	817	1246
Hospital mortality	111 (3.4)	104 (3.4)	91 (5.8)	54 (6.6)	180 (14.5)
Mortality, OR (95% CI)					
Model 1: unadjusted	1 (ref)	0.99 (0.75–1.30)	1.71 (1.28–2.27)	1.97 (1.41–2.75)	4.70 (3.67–6.01)
Model 2 #	1 (ref)	1.04 (0.78–1.38)	1.55 (1.45–2.10)	1.90 (1.32–2.74)	3.23 (2.39–4.38)
Model 3: model 2 and lowest calcium	1 (ref)	1.07 (0.81–1.43)	1.65 (1.22–2.24)	2.04 (1.42–2.94)	3.52 (2.60–4.78)
Model 4: model 2 and highest calcium	1 (ref)	0.99 (0.74–1.31)	1.39 (1.02–1.89)	1.58 (1.09–2.29)	1.93 (1.34–2.79)
Model 5: model 2 and admission calcium *	1 (ref)	1.02 (0.71–1.47)	1.32 (0.90–1.94)	1.76 (1.11–2.79)	2.50 (1.67–3.74)

# Adjusted for age, sex, race, principal diagnosis, Charlson comorbidities score, history of coronary artery disease, congestive heart failure, peripheral artery disease, stroke, diabetes mellitus, chronic obstructive pulmonary disease, cirrhosis, eGFR, ICU admission, acute kidney injury (AKI), the number of serum calcium measurements during hospitalization, and length of stay. * Analysis was limited to patients with available admission calcium

**Table 3 medicina-56-00106-t003:** The association between the direction of serum calcium change and in-hospital mortality.

Calcium Change (mg/dL)	N	In-Hospital Mortality	Model 1	Model 2	Model 3	Model 4	Model 5 *
≤−2.0	768	66 (8.6)	2.72 (1.90–3.88)	2.27 (1.53–3.39)	2.35 (1.58–3.49)	1.32 (0.83–2.10)	1.91 (1.10–3.31)
−1.9 to −1.5	513	21 (4.1)	1.23 (0.75–2.04)	1.31 (0.77–2.20)	1.37 (0.81–2.32)	1.09 (0.64–1.86)	1.51 (0.83–2.74)
−1.4 to −1.0	961	52 (5.4)	1.65 (1.13–2.41)	1.58 (1.07–2.34)	1.65 (1.12–2.45)	1.42 (0.96–2.10)	1.42 (0.88–2.29)
−0.9 to −0.5	1764	55 (3.1)	0.93 (0.64–1.34)	0.96 (0.66–1.40)	0.98 (0.68–1.43)	0.91 (0.62–1.32)	1.06 (0.68–1.66)
−0.4 to 0.0	1885	63 (3.3)	1 (ref)	1 (ref)	1 (ref)	1 (ref)	1 (ref)
0.1 to 0.4	1316	48 (3.7)	1.09 (0.75–1.60)	1.05 (0.71–1.55)	1.07 (0.72–1.58)	1.06 (0.72–1.57)	0.95 (0.57–1.60)
0.5 to 0.9	1266	49 (3.9)	1.16 (0.80–1.70)	1.14 (0.77–1.68)	1.19 (0.81–1.77)	1.11 (0.75–1.65)	0.93 (0.55–1.60)
1.0 to 1.4	614	39 (6.4)	1.96 (1.30–2.96)	1.93 (1.26–2.96)	2.10 (1.36–3.23)	1.82 (1.19–2.80)	1.79 (1.03–3.11)
1.5 to 1.9	304	33 (10.9)	3.52 (2.27–5.47)	3.72 (2.33–5.95)	4.05 (2.52–6.49)	3.23 (2.01–5.20)	3.27 (1.74–6.16)
≥2.0	477	114 (23.9)	9.08 (6.54–12.60)	7.40 (5.08–10.78)	8.44 (5.74–12.41)	5.07 (3.36–7.64)	6.28 (3.82–10.32)

Model 1: Unadjusted; Model 2: Adjusted for age, sex, race, principal diagnosis, Charlson comorbidities score, history of coronary artery disease, congestive heart failure, peripheral artery disease, stroke, diabetes mellitus, chronic obstructive pulmonary disease, cirrhosis, eGFR, ICU admission, AKI, the number of serum calcium measurements during hospitalization, and length of stay; Model 3: model 2 and the lowest serum calcium; Model 4: model 2 and the highest serum calcium; Model 5: model 2 and the admission calcium; * Analysis was limited to patients with available admission calcium. Abbreviations: CI = confidence interval; OR = odds ratio.

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
