# Peer review of "Impact of Changes in Serum Calcium Levels on In-Hospital Mortality"

_medicina, 2020, doi:10.3390/medicina56030106_

Round 1

Reviewer 1 Report

Queries answered to my satisfaction.

Reviewer 2 Report

The revised paper is well organized.
But the originality of the study is a bit disappointing.

Nevertheless, this paper is expected to help people's health.

This manuscript is a resubmission of an earlier submission. The following is a list of the peer review reports and author responses from that submission.

Round 1

Reviewer 1 Report

I carefully reviewed a paper entitled "The Effect of Serum Calcium Level Changes on Hospital Mortality 2".
In general, the data is clear and simple enough to support the conclusion. I carefully checked all the references cited in the manuscript.

These results in this manuscript can be seen in numerous reports.
The author also cited many references (4-6, 10, 11, 15-19).

A small singularity of this manuscript is the effect of changes in serum calcium levels on outcome during hospitalization.

The authors studied the relationship between changes in serum calcium levels during hospitalization and mortality in all hospitalized patients.

However, the results are similar to those in the cited references.

Therefore, this manuscript does not seem to be an original and creative study.

Author Response

Response to Reviewer#1

Comment

I carefully reviewed a paper entitled "The Effect of Serum Calcium Level Changes on Hospital Mortality 2".

In general, the data is clear and simple enough to support the conclusion. I carefully checked all the references cited in the manuscript.

Response: We thank you for reviewing our manuscript and for your critical evaluation.

These results in this manuscript can be seen in numerous reports.

The author also cited many references (4-6, 10, 11, 15-19).

A small singularity of this manuscript is the effect of changes in serum calcium levels on outcome during hospitalization.

The authors studied the relationship between changes in serum calcium levels during hospitalization and mortality in all hospitalized patients.

However, the results are similar to those in the cited references.

Therefore, this manuscript does not seem to be an original and creative study.

Response: We appreciate the reviewer’s input. We agree with the reviewer that previous studies have demonstrated the impact of serum calcium levels, including hypocalcemia and hypercalcemia, on poor clinical outcomes including mortality among hospitalization. However, our study is the first study to assess the association between changes in calcium levels (upward trend and downward trend of serum calcium changes) during hospitalization and mortality. We apologize for being unclear in the original manuscript.

Thus, we have revised manuscript to use the term “upward trend or increase in serum calcium” instead of hypercalcemia. Also, we have revised manuscript to use the term “downward trend or decrease in serum calcium” instead of hypercalcemia, as reviewer’s suggestion.

We also additionally clarified the defitions and analyses in the method and results section throughout manuscript, as reviewer’s suggestion.

We greatly appreciated the editors’ time and comments to improve our manuscript. The manuscript has been improved considerably by the suggested revisions

Reviewer 2 Report

General Comments:  the specific aim of this study is to assess the association between changes in calcium levels during hospitalization and mortality.  In the Introduction the authors highlight the many essential functions of calcium.  The authors provide three references (14, 18, 19) that address a related question.  Akirov (14) reported that abnormal calcium on admission is associated with increased short-term and long-term mortality. They did measure calcium levels before discharge but made no comments on mortality.  In another study (18), hypocalcemia and hypercalcemia on admission were associated with in-hospital mortality. Highest mortality risk is observed in patients with admission hypocalcemia (<7.9 mg/dL). The third study (19), evaluated only hemodialysis patients so it is not relevant for comparison.  Study number 18 was done at Mayo on 18,437 patients whose data were collected between 2009-2013.  The current study uses the same data set but includes only 9568 patients.  I assume the reduced number is because that is the total of patients with 2 or more serum calcium measurements.  Statistical analyses are well described.  Results are clearly presented.  The Discussion is relevant. 

Specific Comments:

  1. How many patients had only two serum calcium levels?When there were 3 or more serum calcium levels how was the rate of change calculated?  In the case of two levels was the delta time similar or close in all patients? 

  1. How many patients had an abnormally high or low serum calcium at any moment during the hospitalization?

  1. How was AKI defined?

  1. Do the results hold when using just the first calcium?Do the results hold when using just the last calcium?  Do the results hold when using only the time averaged serum calcium?

  1. What happens when you add ICU stay at any point during the admission into the model?

Author Response

Response to Reviewer#2

Comment

the specific aim of this study is to assess the association between changes in calcium levels during hospitalization and mortality.  In the Introduction the authors highlight the many essential functions of calcium.  The authors provide three references (14, 18, 19) that address a related question.  Akirov (14) reported that abnormal calcium on admission is associated with increased short-term and long-term mortality. They did measure calcium levels before discharge but made no comments on mortality.  In another study (18), hypocalcemia and hypercalcemia on admission were associated with in-hospital mortality. Highest mortality risk is observed in patients with admission hypocalcemia (<7.9 mg/dL). The third study (19), evaluated only hemodialysis patients so it is not relevant for comparison.  Study number 18 was done at Mayo on 18,437 patients whose data were collected between 2009-2013.  The current study uses the same data set but includes only 9568 patients.  I assume the reduced number is because that is the total of patients with 2 or more serum calcium measurements.  Statistical analyses are well described.  Results are clearly presented.  The Discussion is relevant. 

Response: We thank you for reviewing our manuscript and for your critical evaluation.

Comment #1

How many patients had only two serum calcium levels?When there were 3 or more serum calcium levels how was the rate of change calculated?  In the case of two levels was the delta time similar or close in all patients? 

Response: Thank you very important comment. 3,883 (39%) had only two serum calcium measurements during hospitalization. Among patients with two serum calcium measurements during hospitalization, the median interval between two serum calcium measurements was 1.1 (IQR 0.6-2.2) days.

Regardless of the number of serum calcium measurements, serum calcium change was calculated based on the absolute difference between the maximum and minimum serum calcium values during hospitalization. Our study assessed only the impact of the magnitude and direction of serum calcium change on mortality and did not assess other aspects of serum calcium change such as rate or variability. This is acknowledged in the limitation section.

 Comment #2

How many patients had an abnormally high or low serum calcium at any moment during the hospitalization?

Response: We appreciate the reviewer’s comment. The following statements have been added to the result section.

Given the normal reference range of total serum calcium of 8.6-10.0 mg/dL in our hospital, 1501 (15%) had elevated serum calcium above 10.0 mg/dL, and 6481 (65%) had decreased serum calcium below 8.6 mg/dL during the hospitalization.   

Comment #3

How was AKI defined?

Response: Thank you for important comment. The following statements have been added to the method section to describe the definition of acute kidney injury.

Acute kidney injury (AKI) was defined as an increase in serum creatinine of ≥0.3 mg/dL or ≥1.5 times from the most recent outpatient serum creatinine within in 1 year before admission.

Comment #4

Do the results hold when using just the first calcium?Do the results hold when using just the last calcium?  Do the results hold when using only the time averaged serum calcium?

Response: The association of the first, the last, and the mean serum calcium during hospitalization with in-hospital mortality was shown in Table below. Hypocalcemia (serum calcium ≤8.5 mg/dL) and hypercalcemia (serum calcium ≥10.1 mg/dL) were associated with increased in-hospital mortality, compared with normal serum calcium of 8.6-10.0 mg/dL

First serum calcium (mg/dL)

In-hospital mortality

OR (95% CI)

≤8.5

6.7%

1.67 (1.38-2.01)

8.6-10.1

4.1%

1 (ref)

≥10.1

7.7%

1.93 (1.47-2.54)

Last serum calcium (mg/dL)

In-hospital mortality

OR (95% CI)

≤8.5

5.1%

1.35 (1.11-1.65)

8.6-10.1

3.8%

1 (ref)

≥10.1

27%

9.33 (7.27-11.97)

Mean serum calcium (mg/dL)

In-hospital mortality

OR (95% CI)

≤8.5

5.1%

1.24 (1.02-1.50)

8.6-10.1

4.2%

1 (ref)

≥10.1

17.6%

4.91 (3.83-6.28)

Comment #5

What happens when you add ICU stay at any point during the admission into the model?

Response: The reviewer raised very important point. We additionally adjusted for ICU stay during hospitalization as suggested. Overall, the result is similar to the previous analysis. That is, serum calcium change of ≥1.0 mg/dL was significantly associated with increased in-hospital mortality. We have updated the results in Table 2 and Table 3 as well as the text result section.

We greatly appreciated the editors’ time and comments to improve our manuscript. The manuscript has been improved considerably by the suggested revisions
